# Survey on Sports-Related Concussions among Japanese University Students

**DOI:** 10.3390/brainsci12111557

**Published:** 2022-11-16

**Authors:** Shunya Otsubo, Yutaka Shigemori, Hiroshi Fukushima, Muneyuki Tachihara, Kyosuke Goto, Koki Terada, Rino Tsurusaki, Keita Yamaguchi, Nana Otsuka

**Affiliations:** 1Faculty of Sports and Health Science, Fukuoka University, Fukuoka 814-0180, Japan; 2Graduate School of Sports and Health Science, Fukuoka University, Fukuoka 814-0180, Japan; 3Department of Neurological Surgery, Faculty of Medicine, Fukuoka University, Fukuoka 814-0180, Japan; 4Department of Rehabilitation, Fukuoka University Hospital, Fukuoka 814-0180, Japan

**Keywords:** sports-related concussion, epidemiology, Japan

## Abstract

In recent years, head injuries in sports have garnered attention, and in particular, international discussions have been held on the prevention of and response to sports-related concussions (SRCs). The purpose of this study is to investigate past SRCs experienced by university students in Japan, clarify the state and mechanism of such injuries in each sport, and consider the creation of an environment for future SRC prevention and responses. A questionnaire survey on past SRC experience was conducted among 1731 students who belonged to Fukuoka University in Japan and took “sports medicine” classes in 2020. Responses from 1140 students (collection rate: 65.9%) were obtained. According to this survey, it was revealed that 39 students (3.7%) had experienced SRC. The male–female ratio of those who had experienced SRC was 31 males (79.5%) and 8 females (20.5%). Two males had experienced SRC twice. In this study, SRCs were recognized in a variety of sports, not just in a few contact sports. It is necessary to further disseminate education on head injury prevention and SRCs among both athletes and coaches, because SRCs have been frequently recognized in various sports.

## 1. Introduction

In recent years, head injuries in sports have garnered attention, with the prevention and management of sports-related concussions (SRCs) in particular having become a topic of discussion. Through a joint statement [1,2,3,4,5] by the International Conference on Concussion in Sport, the dangers of SRC have been disseminated internationally as well as in Japan, and The Japan Society of Neurotraumatology and The Japanese Society of Clinical Sports Medicine have pointed out the importance of SRC prevention and early detection [6,7]. For this reason, countermeasures against SRC, such as revising the rules for the purpose of preventing SRC and creating guidelines for recovery from injury, are being implemented, with such efforts mainly being seen in contact sports, such as rugby and soccer [8,9].

A study analyzing data from the National Collegiate Athletic Association (NCAA) in the United States reported that the incidence of SRC was higher in men’s wrestling, men’s ice hockey, women’s ice hockey, and men’s football than in other sports [10]. Furthermore, a survey on SRC limited to contact sports and collision sports reported that most SRCs occurred in men’s football, women’s rugby, men’s ice hockey, men’s wrestling, and women’s soccer [11]. These studies suggest that SRC may occur frequently in sports with a high frequency of contact, thereby also garnering attention as a research subject.

Various studies on SRC in recent years have been conducted on damage to the brain at the time of injury, such as the measurement of head impact using an accelerometer [12] or estimation of brain injury risk by a video analysis [13]. In addition, studies on the mechanism underlying SRC using a baseline evaluation [14] or blood biomarkers are also being conducted [15]. However, most of the published studies on SRC have been conducted on contact sports and other sports. Therefore, rule revisions to deal with SRC in contact sports (American football and rugby) are underway, but little progress has been made in preventing and dealing with SRC in sports that involve little human contact because research is not actively conducted in these sports.

Studies on SRC in Japan have been conducted on athletes in sports with frequent contact, as in other countries [16,17]. While epidemiological studies not limited to a single sport have also been conducted [18], there may be sports events that have not been included in epidemiological studies despite athletes having experienced SRC injuries, as the number of such reports is extremely small. The characteristics of SRC in Japan have not been clarified, as no studies have been conducted comparing the actual state of SRC between Japan and other countries. For these reasons, as a first step in the promotion of SRC prevention and response among the sports associations of Japan, it is necessary to comprehensively investigate the actual conditions of SRC in sports events.

The present study investigated SRCs experienced in the past, clarified the state of SRC injuries and the associated injury mechanisms among different sports and considered the future creation of an environment for SRC prevention and response in Japan.

## 2. Materials and Methods

A questionnaire survey was conducted among 1731 university students who took “sports medicine” classes at Fukuoka University in Japan in 2020. Students in physical education departments (e.g., the Faculty of Sports and Health Science) were not included as subjects. The survey was administered immediately after students attended both basic and more advanced lectures on sports-related concussion as described in SCAT5 and elsewhere.

Because it was difficult to distribute and collect questionnaires due to the impact of COVID-19 in 2020, we conducted a questionnaire survey online using Google Forms. The content of the questionnaire survey included the presence or absence of experience playing sports (including age groups), type of experience playing sports (multiple answers possible), presence or absence of SRC injury experience, number of SRC injury experiences, and presence or absence of attending classes concerning education on SRC injury mechanisms and head injury prevention. In the SRC questionnaire, those who answered “yes” to the question “Have you ever had a SRC?” (e.g., in the third year of elementary school, during a judo match, the athlete was thrown down head first and hit his head, and thus had to go to the hospital because he could not stop vomiting).

Regarding experience playing sports, those who had participated in extracurricular activities or sports clubs were considered to have experience playing sports. Regarding the type of experience playing sports, subjects were asked to include the sports in which they participated through club activities or teams, excluding those engaged in through physical education classes at school.

The SRC injury experience, number of injuries, and mechanism of injury included cases that occurred during physical education classes at school. The subjects were asked to describe in a free-text format the SRC’s mechanism of injury in order to classify the sports category, cause of injury (fall, contact with person, contact with object, unknown), status of injury (during game, practice, unknown), and age at the time of injury (elementary school student, junior high school student, high school student, university student, unknown). Cases for which the sports event could not be specified based on the injury mechanism response, along with responses involving head injuries other than those sustained during sports (falling from playground equipment, etc.), were excluded as targets of this study.

## 3. Results

We were able to obtain responses from 1140 of the 1731 people who took “sports medicine” classes in 2020 (response rate: 65.9%). The respondents included 584 men (51.2%) and 556 women (48.8%).

Of the 1140 respondents, 1045 (91.7%) were experienced athletes, while 95 (8.3%) were inexperienced. Among males (584 respondents), 572 (97.9%) had some sports experience and 12 (2.1%) had no experience, while among females (556 respondents), 473 (85.1%) had some sports experience and 83 (14.9%) had no experience, indicating that males in this study tended to have more experience in playing sports. The proportion of respondents with experience in sports was higher than that of women in this study. A history of SRC was found in 39 of the 1045 experience playing sports (3.7%), for a total of 41 cases. Two of the men had experienced SRC twice. There was a gender difference among SRC, with 31 men (79.5%) and 8 women (20.5%) identified. The prevalence of concussions by gender was 5.4% (31/572) for men and 1.7% (8/473) for women. A total of 59 sports were reported as experienced, including 53 for men and 42 for women (Table 1). SRC occurred in soccer (17 cases), judo (5 cases), baseball (5 cases), kendo (4 cases), basketball (3 cases), rugby (2 cases), volleyball (2 cases), dodgeball (1 case), ice hockey (1 case), and handball (1 case) (Table 2). Of the 41 SRC cases, the number of women injured was 2 in judo, 2 in kendo, 2 in basketball, 1 in volleyball, and 1 in dodgeball. 

The post SRC symptoms were as follows: feeling sick (4 cases), headache (2 cases), loss of consciousness (2 cases), bleeding, lightheadedness, dizziness, nausea, and drowsiness (1 case each), lightheadedness and feeling sick (1 case), bleeding and feeling sick (1 case), headache and feeling sick (1 case), and the number of cases with no description was 23. There were no cases of any residual disability due to SRCs. 

The highest number of SRC injuries was due to falls, with most injuries occurring during games (Table 3). Table 4 shows the injury status by type of sport.

Of the 1140 respondents, 130 (11.4%) had attended classes concerning head injury prevention, while 1010 (88.6%) had no experience therewith. Of the 90 students who had continuously belonged to extracurricular activities or sports clubs from preschool age to university, 18 (20.0%) had attended such classes, while 72 (80.0%) had no experience attending such classes. Of the 39 respondents with a history of SRC injury, 8 (19.5%) had attended, and 31 (79.5%) had never attended such classes.

Of the 41 SRC injuries, 16 (39.0%) visited a hospital, 8 (20.1%) did not, and 17 (41.5%) were unknown. Of the eight respondents (eight cases) who had attended educational classes on head injury prevention, three cases (37.5%) visited the hospital after SRC injury, while five cases (62.5%) were unknown. Of the 31 respondents (33 cases) who had no experience attending such classes, 13 cases (39.4%) had visited a hospital, 8 cases (24.2%) did not visit the hospital, and 12 cases (36.4%) were unknown. In addition, no respondents reported being hospitalized due to SRCs.

Regarding the age group with experience playing sports, the percentage of men who answered that they had experience playing sports was the highest in the junior high school age group response (530 respondents: 90.8%), while women indicated the most frequent age to be from 10 to 12 years of age or from the fourth to sixth year of elementary school (320 respondents: 57.6%) and junior high school (320 respondents: 57.6%) (Table 5) as their most frequent time of experiencing such injuries.

Regarding the age group of SRC occurrence, the study found 3 cases in elementary school students, 14 in junior high school students, 9 in high school students, and 2 in college students, with 13 cases unknown (Table 6).

The types of TBI other than SRC, in addition to intervention and follow-up were not investigated and are unknown, which is one limitation associated with this study.

## 4. Discussion

While studies on SRC in Japan have been conducted on athletes in sports with a high frequency of contact (mainly contact sports) [16,17], there are very few epidemiological studies that are not limited to a single sport [18]. Therefore, the actual state of SRC in Japan is unclear. We therefore conducted a survey of the actual state of SRC experienced by Japanese university students in their past.

It was suggested as a feature of this study that the sports with the highest incidence of SRC in Japan might be different from those in other countries. Zuckerman et al. reported that men’s wrestling, men’s and women’s ice hockey, and men’s football were the top college sports with the highest incidence of SRC [10]. Studies targeting youth and children have reported that SRC occurs frequently in rugby, ice hockey, American football, soccer, and basketball [19,20]. Therefore, the risk of SRC injury is considered high in sports events that involve frequent contact with people, and SRC prevention and countermeasures are being advanced internationally. This study also found that SRC was observed in soccer, basketball, rugby, and ice hockey, which were listed as the sports with the highest incidence of SRC in previous studies. However, rugby and ice hockey, which have been reported to have a high incidence of SRC in previous studies [10,11,19], did not rank high with regard to the number of SRC cases in Japan, whereas judo, baseball, and kendo did rank high, yielding a different result from other countries.

The discrepancy between the present findings and those of previous studies is attributed to there being few people in Japan who have experienced those sports with the highest incidence of SRC in other countries. Among the sports that have been reported as having a high incidence of SRC in other countries, 2 respondents to the present study reported having experienced wrestling, 2 experienced ice hockey, 3 experienced American football, and 14 experienced rugby. In contrast, more than 100 people in total had experienced soccer, baseball, and kendo, which were the sports with the highest number of SRC cases in our study (Table 1). Thus, differences in the population of athletes between Japan and other countries are presumed to be the main reason why the results differed from previous studies conducted overseas, making it clear that the actual conditions of SRC are indeed different between Japan and overseas. Therefore, it is necessary to grasp the actual state of SRC in Japan and consider prevention and countermeasures by focusing on not only sports that have been reported to have a high incidence of SRC overseas but also those that have a large number of participants in Japan, regardless of overseas trends. Ohio State University’s index-based questions are well-known as a method of surveying SRCs among the general public. However, this original survey was conducted on students immediately after they had taken university lectures on the basic knowledge associated with SRC, and this fact may have caused the results to differ from those of previous studies.

Fact-finding surveys on SRC in the world that were not limited to specific sports, as with this study, have rarely reported on SRC occurring in judo, kendo, and dodgeball. However, sports-related head injuries, fatal accidents, and SRC have been confirmed in judo, a sport that originated in Japan [21,22,23]. The present study found that the injury mechanism in judo was due to throwing in all cases, with injuries caused by hitting the head hard. Because a previous study [22,23] also reported that athletes were injured due to throwing techniques with Oosotogari during Randori (freestyle judo training), it is important to prevent SRC by “Ukemi (the art of falling)” when throwing. With kendo, the most frequent injury was due to falling, with one case found in which the head was hit with a bamboo sword. Although studies on SRC in kendo have examined the effects of strikes on the head [24,25], to our knowledge, there has been no research on injuries due to falls. However, in kendo, it is sometimes possible to collide with an opponent and blow him away, therefore suggesting that it is necessary to practice ukemi in kendo as well and that it is important to create rules to prevent falls. 

Martial arts have been a compulsory subject in junior high school physical education programs in Japan since 2012, and the importance of teaching ukemi to prevent head injuries has been pointed out [23]. Therefore, in judo, it is essential to provide sufficient ukemi instructions in order to prevent SRC, especially for beginners and students who practice judo in physical education classes. Although reports of SRC in dodgeball are rare, the main cause of SRC in dodgeball was not identified [26], but in our study it was caused by the ball hitting the head. While it is unclear whether the mechanism of injury in all cases was due to contact with a ball, it was suggested in Japan that SRCs occur even in sports such as dodgeball, which is considered safe in schools (especially for elementary school students) as a childhood game. Therefore, it is necessary for not only sports organizations but also teachers at schools to understand the risk of SRC in such situations. Our study found that most cases of injuries were caused by falls due to various situation, suggesting that it is necessary to practice ukemi in kendo and dodgeball as well and that it is important to create rules to prevent falls.

The male-to-female ratio of SRC victims in this study was 31 men (79.5%) to 8 women (20.5%). The percentage of SRC victims as a whole indicates that the percentage of men with SRC may be high in Japan as well. Previous studies on gender differences in SRC injuries indicated the possibility that women were more likely to suffer SRC because they may not be able to reduce the impact on the head due to the smaller number of muscles in the neck, leading to the belief that women have a higher risk of SRC than men [27,28]. In fact, there are reports that SRC occurs more frequently in women than in men [29]. Furthermore, studies comparing gender differences in SRC incidence have been limited to specific sports, such as soccer and basketball. However, the fact that the percentage of men who participate in sports is clearly higher than that of women is thought to explain the higher rate of men with SRC than women in the present study. Therefore, to compare gender differences in terms of the incidence of SRC in Japanese sports, it is necessary to limit the number of sports, as in overseas studies, and confirm the number of male and female subjects before conducting an analysis.

It became clear through our study that SRC occurred most frequently in junior high school students in Japan. The reasons for this include factors such as an increase in the number of people who have experience in sports, along with an increase in affiliation with sports club activities that involve contact with people. It is believed that one of the factors for this is that the speed of throwing objects and the speed of running have improved due to improvements in basic physical strength, as well as increases in height, weight, and muscle mass with the physical growth of the players, resulting in greater impact force. As mentioned above, it has been suggested that the strength of the neck muscles may affect the impact mitigation of the head [27,28]. It is also possible that many of the injuries occurring in junior high school students were because this is a population at a developmental stage with relatively little muscle mass. Because the difference in physical characteristics is believed to affect the impact force at the time of contact, special attention should be paid to SRC injuries in the junior and senior high school entry years, when differences in physique are likely to occur [22].

This study also revealed that education on head injuries is not widespread in Japan. Twenty years have already passed since 2001, when the International Conference on Concussion in Sport, which is held once every four years, was held for the first time. More and more sports organizations are now working to prevent and respond to SRC. The present study investigated the effect of the presence or absence of attending educational classes on head injury prevention in sports. Despite more than 20 years having passed since SRC initiatives began to be implemented worldwide, 1010 (88.6%) of the 1140 subjects had no experience attending such classes. Of the 90 respondents who had ever belonged to any activities or sports clubs, 72 (80.0%) had no experience attending such classes. In addition, 33 (80.5%) of the 39 who had a history of SRC injuries had never taken such classes. This result suggests that education on the risk of head injuries has not advanced in Japan. 

Because SRC is difficult to judge based on the victim’s appearance alone and there are many aspects that rely on subjective symptoms, in addition to the fact that it is difficult for athletes who are not knowledgeable about SRC to be aware of their own condition, there were likely a certain number of participants who were not aware of ever having had SRC in this study too. Shigemori et al. reported a 9.1% incidence of SRC that athletes were unaware of in a questionnaire survey conducted immediately after the SRC lecture (they such cases call it “unaware SRCs”), thus suggesting that the actual incidence of SRC is higher than that reported in this study [18]. In particular, since SRC tended to occur most frequently in junior high school students in this study, it is considered important to start educating children from a younger age about SRC. Namely, from around 10 to 12 years of age would be the most suitable time to teach children about the dangers associated with SRCs. Japan’s Basic Act on Sport stipulates that the national and local governments must strive to train instructors and provide them with sufficient knowledge regarding safety in order to prevent sports accidents [30]. However, while measures have been taken in contact sports organizations, such as rugby, to educate SRC and develop protocols to safely allow athletes to return to play, SRC education and measures have still not been sufficiently provided in other sports. Therefore, it is important to educate both instructors and athletes about the dangers of SRC in school education (e.g., health and physical education class) and in domestic sports leader training (not limited to contact sports), and the “Concussion Recognition Tool 5” [31] in order to better support non-medical personnel in SRC assessment. Judging from the above, although it is important to educate sports instructors, the spread of education among athletes themselves is also an important issue for the promotion of SRC prevention in Japan.

## 5. Conclusions

As a result of conducting a questionnaire survey among Japanese university students regarding experiences of SRC, a history of SRC was found in 3.7% of the experiences playing sports. Sports with a large number of SRCs included soccer (17 cases), which has a large number of players in Japan, judo (5 cases), baseball (5 cases), and kendo (4 cases), revealing differences in SRC rates among sports events and in characteristics of sports in which SRC is likely to occur between Japan and other countries. In addition, since SRCs in Japan occur most frequently among junior high school students, children should be taught about SRCs at a young age, namely before they enter junior high school. Furthermore, because there may exist sports organizations other than for contact sports that require the active implementation of preventive measures, it is important to continue comprehensive epidemiological surveys, in addition to epidemiological surveys limited to specific events. Because the present study revealed that there are many people in Japan who have never taken educational classes on sports-related head injuries, it is important to disseminate education on the prevention of and response to SRC, not only among instructors and coaches involved in sports but also among school officials.

## Figures and Tables

**Table 1 brainsci-12-01557-t001:** Type of experience playing sports in the past.

No	Sports	Pre-School Child		The 1st to the 3rd Grade of Elementary School		The 4th to the 6th Grade of Elementary School		Junior High School		High School		University		Total
		Male	Female	Male	Female	Male	Female	Male	Female	Male	Female	Male	Female	
1	Swimming	123	115	184	143	141	110	22	16	4	6	3		867
2	Soccer	99	13	126	7	139	9	116	4	77	6	19	0	615
3	Baseball	39	0	84	1	112	1	131	0	88	0	26	0	482
4	Basketball	11	10	19	18	46	36	65	36	40	14	13	6	314
5	Tennis	4	6	9	13	21	29	68	71	47	31	9	5	313
6	Athletics	7	4	14	11	20	16	48	43	35	14	3	1	216
7	Volleyball	4	4	8	22	9	36	16	35	19	14	3	6	176
8	Badminton	3	8	2	10	7	39	10	29	16	23	8	8	163
9	Table tennis	3		3	5	7	10	36	40	20	9	3	4	140
10	Karate	19	5	30	11	28	11	9	2	5		3		123
11	Softball	10		38	1	48	3	1	12	1	4	2		120
12	Kendo	7	3	13	5	17	7	20	15	11	7	3	2	110
13	Dance		11		21		24		6	1	24	2	12	101
14	Gymnastics artistic	13	29	5	12	4	9		2		1			75
15	Rhythmic gymnastics	1	16		18		14		4		4			57
16	Ballet		17		14		13		7		3		2	56
17	Handball	1	1	2	1	3	4	6	6	11	7	2	1	45
18	Kyudo	1	1					3		12	21	1	1	40
19	Judo	2	1	6	4	5	4	8	2	3	1	1		37
20	Dodgeball	2	1	6	1	8	3	1						22
21	Baton twirling		4		4		4		4		3			19
22	Shorinji kempo			3	2	3	3		2	1		1	1	16
23	Golf	1	1	3	1	3	2		2	1	1		1	16
24	Rugby	3	1	3		1		1		4	1			14
25	Aikido	1	1		2	1	2		1	1	1	1	1	12
26	Squash	0	0	0	0	0	0	0	0	0	0	5	6	11
27	Skating	1	3	1	2		4							11
28	Skiing	2		1		1		1		1		2	2	10
29	Lacrosse											4	4	8
30	Naginata							1	2		3		1	7
31	Wrestling	1		1	1		1		1		1		1	7
32	Cheerleading												6	6
33	Snowboard	1		1		1		1		1		1		6
34	Kickboxing									2		1	1	4
35	Sumo			1			1	1	1					4
36	Art of self-defense	1		1		1		1						4
37	Taekwon-do		1	2		1								4
38	American football											3		3
39	Mountaineering									2		1		3
40	Ice Hockey											2		2
41	Other (19 types)	0	2	3	0	4	2	0	0	5	2	10	2	30

**Table 2 brainsci-12-01557-t002:** Number of SRC occurrences by type of sport (*n* = 41).

Sports	Number	Total Number of Players	Rate of SRC (SRCs/Total Number of Players)
Soccer	17	615	2.8%
Judo	5	37	13.5%
Baseball	5	482	1.0%
Kendo	4	110	3.6%
Basketball	3	314	1.0%
Rugby	2	14	14.3%
Volleyball	2	176	1.1%
Ice Hockey	1	2	50.0%
Dodgeball	1	22	4.5%
Handball	1	45	2.2%

**Table 3 brainsci-12-01557-t003:** Occurrence of SRC.

Situation Caused by an Injury	Game	Practicing	Unknown	Total
Fall	9	3	5	17
Contact with person	7	2	2	11
Contact with objects	3	3	5	11
Unknown	2	0	0	2

**Table 4 brainsci-12-01557-t004:** Occurrence of SRC by type of sport.

Sports	Person-to-Person Contact	Contact with Objects	Fall	Unknown
Soccer	7	2	6	2
Judo	0	0	5	0
Baseball	1	3	1	0
Kendo	0	1	3	0
Basketball	1	1	1	0
Rugby	1	0	1	0
Volleyball	0	2	0	0
Dodgeball	0	1	0	0
Ice Hockey	1	0	0	0
Handball	0	1	0	0

**Table 5 brainsci-12-01557-t005:** Number of people with experience playing sports by age group.

	Male (*n* = 584)	Female (*n* = 556)
Pre-school child	269 (46.1%)	206 (37.1%)
1st to 3rd grade of elementary school	418 (71.6%)	278 (50.0%)
4th to 6th grade of elementary school	472 (80.8%)	320 (57.6%)
Junior high school	530 (90.8%)	320 (57.6%)
High school	405 (69.3%)	203 (36.5%)
University	131 (22.4%)	80 (14.4%)

**Table 6 brainsci-12-01557-t006:** Number of people experience SRC by age group.

Pre-School Child	Elementary School	Junior High School	High School	University	Unknown
0	3	14	9	2	13

## Data Availability

Not applicable.

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
