# Peer review of "Survey on Sports-Related Concussions among Japanese University Students"

_brainsci, 2022, doi:10.3390/brainsci12111557_

Round 1

Reviewer 1 Report

Thank you for the opportunity to review this manuscript.  Overall, this is an excellent study and I believe it to be an important addition to the literature.  I have one major concern that should be addressed, as well as several minor textual comments.

Main concerns
- In the methods section, I would provide the prompt used to determine the number of sports related concussions.  This is important because prior research has indicated that providing a definition of what constitutes a concussion significantly increases the number of reported events.  It appears that many people do not have a good understanding of what constitutes a concussion.  This may be of particular concern in this study, as it appears that education on concussion within this population is limited.

- If participants were not provided a definition of what constitutes a concussion, then you may want to clarify in the discussion that the reported rates may be lower due to this factor.  A good place for expanding on this might be around lines 247-248, where this issue is briefly addressed.  

Minor Concerns/edits:

- On line 45 I would delete the word "events."
- I had trouble understanding lines 51-54.  On line #52, I'm not sure the part about "fatal head injuries" is necessary.  At this point there have been studies on sports-related concussions in a wide variety of sports.  Additionally, I would re-work lines 52 through 54 to improve clarity. 
- Around line #77 might be a good place to include the question that was used to prompt for the number of sports-related concussion experiences (see comment above for more details).
- Lines 101 to 104 have some minor grammatical errors and could be re-written to improve clarity. 
- On line #119 I might consider removing the word "survivors," given that concussions are mild traumatic brain injuries.
- On line #164, the discrepancy may also be due to how the question about sports-related concussions was asked (see comment above).  This also applies to lines 171 through 173.  You may want to consider including this possibility within this section by adding a sentence or two.
- On line #190, the first word of the second sentence should start with a capital "T."
- On line #204, a word is missing after "various."

Again, that you for the opportunity to review this manuscript.  This is an excellent and important study. 

Reviewer 2 Report

Observational study of 1140 students in Japanese University who took sports medicine classes and were surveyed regarding history of sport-related concussion. The authors found that 3.4% of surveyed students had concussions (N=39). 

While i do not have major concerns regarding the data, the manuscript remains simple and descriptive, the survey method for asking about sport-related concussion did not utilize an objective assessment tool (e.g. the Ohio State University Traumatic Brain Injury Identification Method), and there were no interventions provided nor imminent next steps indicated. From an epidemiological perspective, the study would be of greater interest and utility if it reported other forms of TBI beyond concussion, rates of hospitalization, postinjury deficits, interventions, follow-up, and/or whether concussion education was provided. As the manuscript currently stands, I am struggling to find relevant take-aways.

Reviewer 3 Report

This paper reports on results of a sports concussion related questionnaire completed by University students in Japan. I do think the argument that SRCs are often only studied within identified high incidence sport and therefore our understanding of SRCs may be a bit limited is reasonable, and the aims to get a broader understanding of how and when SRCs occur is valid. However, the presentation of the results are a bit difficult to follow and limit enthusiasm for the paper.

-          Important information like the number/percentage of respondents who had participated/played a sport in their lifetime was not directly reported – this is an important number as this is what should be the denominator when determining rates of SRC. If a respondent has no history of sports participation then they would be unable to sustain an SRC.

-          Starting the results section with the break down by age group feels out of place. I think it would be easier to follow if the authors report overall numbers (i.e., number/proportion of respondents who played sports, number/proportion with history of SRC, breakdown by sports, and then the rate of SRCs based on participants of each sports). Then moving to talking about the idea that kids play sports as early as pre-school and sustain SRCs as early as elementary school is valid, but needs to come later in the results section to help reader orient themselves to the data a bit more.

-          The authors do report a number of tables with count data that are not very helpful – particularly because more than 1000 students completed the survey. This is true throughout all of the tables, but especially table 4 when the authors are presenting number of SRCs by sport. While soccer has the greatest number of SRCs – it is also likely the most popular sport played. In contrast, Kendo has fewer SRCs, but also likely has much fewer participants, and therefore the rate of SRC in Kendo may actually be much higher than soccer. A more compelling statistic would be to report the number of SRCs in soccer/number of soccer participants, etc.

-          The large table with all sports and then counts of participants by age and sex is a bit overwhelming. There are no totals and other than showing the wide range of activities it seems of limited value.

-          Along these lines – I am sure that there are kids who have participated in more than 1 sport during their lifetime, but the breakdown by age group and/or sex makes it difficult for the reader to make conclusions about this.

Discussion

-          In general, I think the author’s point that much of the SRC literature is focused on American football and ice hockey that may not be common sports in Japan and therefore this is a need for this study is compelling, but again the way the data is presented in the results make this hard for the reader.

-          Again the information that participants start playing sports as early as preschool with concussions starting as early as elementary school is a compelling argument for SRC education to begin early in life but this gets a bit bogged down in results section.

-          Authors present some stats in the discussion section for the first time (i.e. male to female ration of SRC victims) – this would be helpful in the results section. Authors bring up the idea that this may be due to males playing more sports, but they don’t look at this in their own data – they could very easily look at the rate of SRCs in the sport playing males vs. females in their sample.

-          Again, in the conclusion section of the paper, authors report the proportion of SRCs that occur in each sport – this info should be included in the results section, but as is – the ratio of soccer srcs/total srcs is a bit misleading given the high rate of soccer participation. I think it would be more vauable/compelling to report the rate of SRCs in each sport (within soccer, #SRCs/#soccer players – for all sports).

-          A final minor note – throughout the paper authors use the phrase ‘experience sports’ but participated in sports or experience playing sports may be a more appropriate term.

Round 2

Reviewer 2 Report

I appreciate the authors' efforts to improve the relevance of their manuscript. While the study is observational, it now includes points which highlight the challenges of obtaining sports-related concussion data in young adults.

I would recommend that the authors add supporting evidence and guidance on improving awareness of SRCs in the parental and young adult community for primary prevention, and how to seek medical attention if injuries occur at a younger age. 
